# Effects of Pegylated Interferon Alpha and Ribavirin (pegIFN-α/RBV) Therapeutic Approach on Regulatory T Cells in HCV-Monoinfected and HCV/HIV-Coinfected Patients

**DOI:** 10.3390/v13081448

**Published:** 2021-07-25

**Authors:** Kamil Grubczak, Anna Grzeszczuk, Monika Groth, Anna Hryniewicz, Anna Kretowska-Grunwald, Robert Flisiak, Marcin Moniuszko

**Affiliations:** 1Department of Regenerative Medicine and Immune Regulation, Medical University of Bialystok, 15-269 Białystok, Poland; anna.kretowska-grunwald@umb.edu.pl; 2Department of Infectious Diseases and Neuroinfections, Medical University of Bialystok, 15-540 Bialystok, Poland; neuroin@umb.edu.pl; 3Department of Allergology and Internal Medicine, Medical University of Bialystok, 15-089 Bialystok, Poland; monika.groth@umb.edu.pl; 4Department of Rehabilitation, Medical University of Bialystok, 15-089 Bialystok, Poland; rehab@umb.edu.pl; 5Department of Infectious Diseases and Hepatology, Medical University of Bialystok, 15-540 Bialystok, Poland; robert.flisiak@umb.edu.pl

**Keywords:** pegIFN-α/RBV, HCV, HIV, HIV/HCV-coinfection, regulatory T cells

## Abstract

Approximately 25% of HIV-infected patients are co-infected with HCV. Notably, the burden of HCV infection (e.g., viral persistence, viral load, or HCV-related liver symptoms) is more pronounced in the presence of HIV co-infection. However, to date, the underlying immune mechanisms accounting for accelerated disease progression in HIV/HCV-coinfected individuals have not been described in sufficient detail. We hypothesized that regulatory T cells (Treg) bearing potent immunosuppressive capacities could not only play a substantial role in the pathogenesis of HCV/HIV coinfection but also modulate the response to the standard anti-viral therapy. Materials and Methods: To this end, we studied alterations in frequencies of Treg cells in correlation with other Treg-related and virus-related parameters in both HCV and HCV/HIV-infected patients subjected to standard pegIFN-α/RBV therapy. Results: Notably, we found that pegIFN-α/RBV therapy significantly increased levels of Treg cells in HCV-infected but not in HIV/HCV-coinfected individuals. Furthermore, HIV/HCV-coinfection was demonstrated to inhibit expansion of regulatory T cells during anti-viral treatment; thus, it might probably be responsible for viral persistence and HCV-related liver damage. Conclusions: Therapy with pegIFN-α/RBV demonstrated a significant effect on regulatory T cells in the course of HIV and/or HCV infection indicating a crucial role in the anti-viral immune response.

## 1. Introduction

Hepatitis C virus (HCV) and human immunodeficiency virus (HIV) coinfection affect approximately 25% of HIV-infected individuals. HIV coinfection constitutes around 3–5% of all HCV-positive patients [1]. Clinical evidence has shown a higher rate of HCV viral persistence and increased viral load in HCV/HIV-coinfected patients compared to subjects with HCV monoinfection [2]. HIV infection is also associated with higher HCV RNA viral load and more rapid progression of HCV-associated liver diseases [3,4,5].

Mechanisms of accelerated disease progression in HIV/HCV-coinfection have not been fully determined. However, disturbances in virus-specific T cell responses via altered activity of regulatory T cells (T_reg_) are thought to play an important role [6,7,8] in infections caused by HIV, HCV, and herpes virus [9,10,11,12]. Prior work has shown T_reg_ expansion in the course of HIV and/or HCV infection [13,14,15]. Notably, several studies have indicated that HIV/HCV coinfection may modify the immunomodulatory activities of these viruses as compared to monoinfection [16,17].

In both HCV-monoinfected and HCV/HIV-coinfected individuals, T_reg_ and other factors responsible for the induction of suppressive cells were found to play a crucial role in modulating HCV-specific immune responses. T_regs_ are thought to exert their deleterious effects through excessive suppression of effector T cells in the antiviral response [18,19]. Approaches aimed at blocking the activity of T cells with a regulatory phenotype may improve antiviral immune responses [20] and protect from HCV-related complications such as liver fibrosis [21]. In this study, we focused on a T_reg_ phenotype defined as CD4^+^CD25^+^CD127^−^Foxp3^+^; however, considering the high correlation of CD127 with Foxp3, we also distinguished the CD4^+^CD25^+^CD127^−^ population as putative T_regs_ [22]. These phenotypes are mainly associated with immunosuppresive functions demonstrated by interaction with dendritic cells, the inhibition of effector lymphocytes, and the release of factors like IL-10, TGF-beta, and IL-35 [23].

Standard therapy in chronic HCV infection involved combined pegylated interferon alpha (pegIFN-α) and ribavirin (RBV). Notably, prior work has indicated that pegIFN-α/RBV application in HCV/HIV-coinfected patients results in sustained HCV eradication in less than 40% of individuals [1]. Despite the introduction of direct-acting antivirals (DAA) in chronic HCV treatment, the established combined pegIFN-α/RBV therapy is still relevant [24,25], especially in regions where there is a substantial risk of HIV coinfection [26].

Therefore, notwithstanding the beneficial therapeutic effects of pegIFN-α/RBV versus standard IFN-α/RBV, further studies are needed to thoroughly evaluate pegIFN-α/RBV effectiveness and assess its utility in HCV-infected patients [27].

In this study, we aimed to assess the effects of combined pegIFN-α/RBV therapy on regulatory T cells and their association with immune system status and viral load in cohorts of HCV-infected and HCV/HIV-coinfected participants. In addition, we evaluated the effects of HIV/ HCV-coinfection on the suppressive activity of T cells and the anti-viral response.

## 2. Materials and Methods

### 2.1. Patients

In the experiment, we included three study groups: HCV-infected patients (*n* = 11; 55% male), HCV/HIV-coinfected patients (*n* = 16; 69% male), and healthy individuals (*n* = 18; 56% male) as a control group.

Diagnosis of chronic HCV infection was defined as the presence of HCV RNA for more than 6 months measured by real-time PCR (Abbott RealTime PHCV, Champaign, IL, USA) (Lower Limit of Quantitation—LLQ, for HCV RNA was 12 IU/mL) and positive results of anti-HCV antibodies measured by a third-generation enzyme immunoassay (Vitros ECI^®^: Ortho-Clinical Diagnostics, Rochester, NY, USA). HCV genotyping was evaluated using a commercial assay (Abbott RealTime HCV Genotype II). Plasma HIV RNA was measured using Versant HIV-1 RNA version 3 (Simens, Barcelona, Spain) (the limit of detection (LOD) for HIV RNA was 5 HIV-1 copies/mL). All patients were negative for hepatitis B surface antigen. In addition, peripheral blood samples were collected from patients before and after 6–12 months of anti-viral therapy consisting of pegylated interferon plus ribavirin (pegIFN-α/RBV). Treatment included one dose of 180 mg pegylated interferon alpha (pegIFN-α) weekly (Pegasys^®^; Hoffman-La Roche, Basel, Switzerland), combined with 1000mg of ribavirin (RBV) daily (Copegus^®^; Hoffman-La Roche). HAART was maintained in subjects with HIV-coinfection; thus, effective control of its viral load was reported in patients enrolled in the study (Appendix A). Patients were recruited at the consultation point for HIV-infected and AIDS patients at the Department of Infectious Diseases and Hepatology, Medical University of Bialystok (Poland). The study was approved by the Local Ethics Committee, and written informed consent was obtained from all patients.

### 2.2. Peripheral Blood Mononuclear Cells (PBMC) Isolation

Peripheral blood mononuclear cells (PBMC) were isolated by density gradient centrifugation using Ficoll (Amersham Biosciences, Piscataway, NJ, USA). Following isolation of the PBMC layer, two washing steps were performed with phosphate-buffered saline (PBS, Biomed Lublin, Poland). Viable cells were cryopreserved in fetal bovine serum (FBS, Gibco, Thermo Fisher, Carlsbad, CA, USA) with 10% dimethyl sulfoxide (DMSO, Sigma-Aldrich, St. Louis, MO, USA) and stored in liquid nitrogen until flow cytometric analyses.

### 2.3. Flow Cytometric Analysis

Thawed peripheral blood mononuclear cells (approximately 0.5 × 10^6^ cells) were stained with fluorochrome-conjugated monoclonal antibodies (5 µL of each antibody) including: anti-CD^4^FITC (clone RPA-T4), anti-CD25 PE-Cy5 (clone M-A251), and anti-CD127 AlexaFluor647 (clone HIL-7R-M21) (BD Bioscience, San Jose, CA, USA). Samples were incubated for 30 min, at 4 °C, in the dark, and then washed in PBS. Subsequently, cells were subjected to a 10-min permeabilization, in the dark, at room temperature (FACS Permeabilizing Solution 2; BD Bioscience, San Jose, CA, USA) to allow staining of intracellular markers. Samples were then washed with PBS and stained with 20 µL of anti-Foxp3 PE (clone 259D/C7) (BD Bioscience, San Jose, CA, USA) monoclonal antibodies and incubated 30 min, at 4 °C, in the dark. Following incubation with antibodies, cells were washed in PBS and fixed (CellFIX; BD Bioscience, San Jose, CA USA). Flow cytometric data were acquired on FACS Calibur flow cytometer (BD Bioscience, San Jose, CA, USA) and analyzed using FlowJo (Tree Star, Ashland, OR, USA). CD4^+^ T lymphocytes were identified using a double-gating strategy of forward- and side-scatter identification in combination with high cellular CD^4^expression. Foxp3-based identification of T_regs_ was performed within CD4^+^ lymphocytes and furthermore within more specific Treg population described as CD4^+^CD25^+^CD127^−^ cells (putative Tregs). The gating strategy was performed on the basis of unstained, FMO (fluorescence-minus-one), and negative population controls (Appendix A).

### 2.4. Statistical Analysis

Statistical analysis was performed using GraphPad Prism 5.0 (GraphPad Software Inc., La Jolla, CA, USA). Assessment of the Gaussian data distribution was estimated with the use of Shapiro–Wilk and D’Agostino–Pearson normality tests. The normality distribution required confirmation with two independent tests, otherwise data were further tested with non-parametric tests. The Mann–Whitney test was used to compare healthy individuals, HCV-infected, and HCV/HIV-coinfected patients. The analysis of differences before and after treatment was performed within each cohort using the Wilcoxon signed-rank test. Because of the lack of a Gaussian data distribution, mutual correlations between tested parameters were analyzed with the use of the nonparametric Spearman correlation test. Results are presented within tables and graphs as median values with 25th and 75th percentiles.

## 3. Results

Characteristics of the studied subjects based on immunological, biochemical, and virological laboratory results.

No significant differences were observed in routine laboratory tests results between the control group, HCV-infected, and HCV/HIV-coinfected cohorts prior to pegIFN-α/RBV-based treatment. HCV-infected patients were clinically stable with no critical changes in, inter alia, liver function-related parameters, with only a slight increase in serum alanine aminotransferase (ALT) and gamma-glutamyl transferase (GGT). Furthermore, HCV/HIV-coinfected patients demonstrated adequate treatment-induced control of HIV viral load with a favorable level of CD4^+^ T cells in the blood and marginally elevated serum aspartate aminotransferase (AST), ALT, and GGT levels. Patients infected with HCV showed a significantly lower score for the AST-to-platelet ratio index (APRI) compared to HCV/HIV-coinfected patients, suggesting a higher risk of hepatic fibrosis in the latter group of patients. Moreover, HCV/HIV-coinfected patients had a lower HCV viral load compared to subjects with HCV-monoinfection (Appendix A).

### 3.1. Alterations in Frequencies and Absolute Numbers of Regulatory T Cells in HCV-Infected and HCV/HIV-Coinfected Patients in the Course of Anti-Retroviral Therapy

We found that both HCV-infected and HCV/HIV-coinfected patients had significantly higher frequencies of regulatory cells, namely, CD4^+^Foxp3^+^ and CD4^+^CD25^+^CD127^−^Foxp3^+^ T cells, compared to the control group. Interestingly, standard antiviral treatment resulted in a significant increase in these populations in patients with HCV infection, whereas no changes were observed in the HCV/HIV-coinfected group (Figure 1a,c). Comparable alterations were found in the suppressive potential within CD4^+^CD25^+^CD127^−^ cells based on Foxp3 expression (Figure 1b). We did not observe any changes in the frequency of putative CD4^+^CD25^+^CD127^−^ regulatory T cells in HCV-infected patients (Figure 1d) (Appendix A).

Analysis of CD4^+^CD25^+^CD127^−^Foxp3^+^ cells confirmed that HCV-infected and HCV/HIV-coinfected patients have a greater number of regulatory cells. However, no differences were found in reference to CD4^+^Foxp3^+^ cells. Furthermore, antiretroviral therapy led to a decline in CD4^+^Foxp3^+^ cell counts in HCV/HIV-coinfected patients, and unlike frequencies, absolute numbers of CD4^+^CD25^+^CD127^−^Foxp3^+^ T cells remained unchanged in the course of treatment (Figure 2a,b). Similarly, changes in putative CD4^+^CD25^+^CD127^−^ cells were in accordance with CD4^+^CD25^+^CD127^−^Foxp3^+^ cell number, however, with significantly lower levels following antiretroviral therapy in both groups (Figure 2c). Interestingly, we found that pegIFN-α/RBV administration in HCV-infected patients led to a slight decrease in the total number of CD4^+^ T cells. In HCV/HIV-coinfected patients, antiviral therapy exacerbated the expansion of CD4^+^ lymphocytes (Figure 2d) (Appendix A).

### 3.2. Effects of Anti-Viral Therapy on T Cell Activation- (CD25) and Development-Associated Protein (CD127) in HCV-Infected and HCV/HIV-Coinfected Patients

We did not find any significant differences in the frequency of CD25^+^ T cells in the experimental groups compared to the control, with only a slight increase in HCV/HIV-coinfected patients. However, mean fluorescent intensity of CD25 was greater following antiretroviral drug exposure in HCV-infected and HCV/HIV-coinfected individuals (Figure 3a,b). In reference to CD127 expression, both frequency and MFI were lower in patients with monoinfection or coinfected with HIV before treatment and remained unchanged in response to treatment (Figure 3c,d) (Appendix A).

### 3.3. Variations in Regulatory T Cells Frequencies and Absolute Numbers among HCV-Infected and HCV/HIV-Coinfected Patients Considering Genotype of Hepatitis C Virus

Considering the different genotypes of the hepatitis C virus, there were no observable differences in CD4^+^Foxp3^+^ phenotype in terms of both frequency and absolute cell count (Figure 4a,e). Furthermore, analysis of CD4^+^CD25^+^CD127^−^Foxp3^+^ cells indicated that patients infected with HCV types 1 and 3 demonstrate an increased absolute number of these cells. This was additionally noted in all three studied HCV genotypes of HCV/HIV-coinfected individuals. Frequencies of CD4^+^CD25^+^CD127^−^Foxp3^+^ seem to correspond to these results; however, a significant difference was observed only in HCV/HIV-coinfected patients with HCV type 3 (Figure 4c,f). Similarly, the frequency of immunosuppressive Foxp3^+^ phenotype within regulatory T cells was found to be higher only in coinfected patients with the HCV type 4 genotype (Figure 4b). In reference to the putative regulatory CD4^+^CD25^+^CD127^−^ phenotype, results were in accordance with absolute numbers of CD4^+^CD25^+^CD127^−^Foxp3^+^ cells, with a lack of difference only in HCV/HIV-coinfected individuals with the HCV type 3 infection. Frequency analysis of CD4^+^CD25^+^CD127^−^ cells, however, showed higher values only in HCV type 3 patients with coinfection (Figure 4d,g). In addition, we demonstrated significantly lower absolute numbers of CD4^+^ cells in HCV-infected patients with HCV type 1 and HCV type 3 in HCV/HIV-coinfected subjects (Figure 4h).

### 3.4. Hepatitis C Virus Genotype Influence on Differential Response of Regulatory T Cells to Anti-Viral Treatment of HCV-Infected and HCV/HIV-Coinfected Patients

When analyzing the effects of treatment on frequencies of CD4^+^Foxp3^+^ cells, we found increased values only in HCV-infected patients with HCV types 1 and 3 (Figure 5a). However, individuals coinfected with HCV types 1 or 3 and HIV demonstrated even lower levels of absolute CD4^+^Foxp3^+^ cell numbers in response to antiretroviral therapy (Figure 6a). In reference to the CD4^+^CD25^+^CD127^−^Foxp3^+^ phenotype, a similar data distribution was observed compared to CD4^+^Foxp3^+^ frequencies in HCV-infected patients. Moreover, here we also demonstrated higher values, which remained unchanged despite treatment, of these cells in HCV/HIV-coinfected subjects with HCV type 3 (Figure 5c). Considering absolute numbers of CD4^+^CD25^+^CD127^−^Foxp3^+^ cells, patients infected with HCV types 1 and 3 only showed higher initial levels of these cells and did not respond to this therapeutic approach. In HCV/HIV-coinfected individuals, all three types of HCV demonstrated higher values of the regulatory phenotype, with a presumed decline in patients with HCV types 1 and 4 following antiretroviral therapy (Figure 6b). Interestingly, treatment application in HCV-infected subjects led to an increase in the immunosuppressive Foxp3^+^ phenotype within CD4^+^CD25^+^CD127^−^ cells in patients with HCV types 1, 3, and probably 4 (despite a lack of significance associated with the small sample size). In reference to HCV/HIV-coinfected patients, significantly higher values in HCV type 4 patients remained unchanged in response to anti-retroviral therapy (Figure 5b). No significant differences were found in HCV-infected individuals in regards to CD4^+^CD25^+^CD127^−^ phenotype frequencies, with higher levels not responding to treatment in HCV/HIV-coinfected patients with HCV type 3 (Figure 5d). Absolute CD4^+^CD25^+^CD127^−^ cell number analysis revealed increased values in HCV-infected patients with HCV types 1 and 3, with only type 1 responding to therapy, and a non-significant decline in those infected with HCV type 4. In HCV/HIV-coinfected subjects, individuals infected with HCV types 1 and 4 demonstrated higher levels of putative regulatory T cells, which decreased (albeit insignificantly) following antiretroviral therapy (Figure 6c). The therapeutic approach seemed to have caused a crucial decline in total CD4^+^ cell number only in patients infected with HCV type 4, both in HCV-infected and HCV/HIV-coinfected individuals (Figure 6d).

### 3.5. Association between Regulatory T Cells and Immunological, Biochemical and Virological Laboratory Parameters, in HCV-Infected and HCV/HIV-Coinfected Patients

In HCV-infected patients, we found a negative correlation in CD4^+^Foxp3^+^ absolute cell number and HCV viral load prior to treatment. This population of cells, as well as the absolute number of CD4^+^CD25^+^CD127^−^Foxp3^+^ cells, demonstrated a positive correlation with serum CD4^+^ lymphocyte levels. The frequency of putative CD4^+^CD25^+^CD127^−^ T cells was found to correlate negatively with CD127 expression within the CD4^+^ cell population. No correlations were observed between regulatory T cells and biochemical parameters related to liver damage, namely, AST, ALT and GGT. When analyzing association between treatment-related change in Treg parameters and other factors, correlations with HGB, CD4^+^, and CD4^+^CD25^+^ cells level were maintained. In addition, changes in Treg-associated parameters correlated significantly with AST, ALT, and GGT (Supplementary Figure 2).

The numbers of CD4^+^Foxp3^+^ regulatory T cells were found to correlate positively with the HIV viral load in untreated HCV/HIV-coinfected subjects. As regards the HCV RNA level, the frequency of Foxp3^+^ cells within the CD4^+^CD25^+^CD127^−^ population showed a positive correlation; however, the absolute number of putative CD4^+^CD25^+^CD127^−^ regulatory T cells demonstrated a negative correlation with the HCV viral load. Similarly to HCV-infected patients, a regulatory T cell-related parameter, absolute CD4^+^CD25^+^CD127^−^ cell number, correlated positively with levels of CD4^+^ T cells in HCV/HIV-coinfected individuals. Additionally, a negative correlation was found in relation to the frequency of CD4^+^CD25^+^CD127^−^Foxp3^+^ and putative CD4^+^CD25^+^CD127^−^ cells. Despite a lack of correlation between regulatory T cells and AST, ALT, or GGT, we observed a positive link between putative CD4^+^CD25^+^CD127^−^ regulatory cells and the parameter associated with hepatic fibrosis—APRI. Furthermore, in the context of change in Treg-related parameters and their correlation with other studied factors, associations have been maintained in reference to PLT, CD4^+^, and CD4^+^CD25^+^ cells. As observed in HCV patients, a correlation with ALT also appeared (Appendix A).

## 4. Discussion

Recent data indicate that HIV/HCV-coinfection influences immune system activity during pegIFN-α/RBV treatment by inducing higher HCV-specific T cells responses in approaches aimed at reducing regulatory T cells activity [20]. Work by Rallon et al. indicated that the frequency of regulatory CD4^+^ T cells, namely, Foxp3^+^ and CD25^+^Foxp3^+^, was only increased in HIV-infected and HCV/HCV-coinfected patients [28]. Here, however, we have demonstrated using more precise immunophenotyping (CD4^+^CD25^+^CD127^−^Foxp3^+^) methods and absolute number counts that increased values of T_regs_ are not limited to infection with HIV but are present in HCV-infected and HCV/HIV-coinfected patients as well. Our data are consistent with results that demonstrated higher levels of regulatory cells present in patients infected with HCV [29,30,31]. Similar to prior work indicating a lack of effect of HCV-coinfection on T_regs_ frequencies in HIV-infected individuals [13], we found no differences in regulatory T cells between HCV-infected and HCV/HIV-coinfected patients. Interestingly, analysis of hepatic T_regs_ revealed higher numbers of these cells in patients with HCV monoinfection, suggesting a critical role of T cell imbalance in controlling excessive immune responses within tissues [32]. 

Here, we found that application of combined pegylated interferon and ribavirin (pegIFN-α/RBV) led to markedly higher frequencies of T cells, demonstrating a suppressive phenotype in HCV-infected patients, as opposed to the lack of treatment effects in HCV/HIV-coinfection. It is worth noting that such differences were not observed when analyzing absolute numbers of these cells. However, we demonstrated that the presence of HIV in HCV-infected patients was associated with a significant decline in total regulatory CD4^+^Foxp3^+^ T cells in response to pegIFN-α/RBV treatment. In addition, the decline in the proportion of regulatory T cells during HAART therapy, based on a less detailed analysis of CD4^+^CD25^+^Foxp3^+^ (without using the CD127 marker), was demonstrated in recent years [13]. Despite the presumed deleterious effects of T_reg_ activity in HIV and HCV infection [18,33], it was found that in antiviral responses, these cells may exert favorable effects by suppressing excessive T cell activity and protecting against reinfection [34]. Therefore, the observed negative correlation between the absolute number of regulatory CD4^+^Foxp3^+^ cells and HCV viral load observed in the present study supports that hypothesis. Therefore, the increased T_regs_ frequency in HCV-infected and HCV/HIV-coinfected patients, even following treatment, may be regarded as a favorable effect in controlling activity of effector T cells and viremia. Further studies should investigate the role of T_regs_ in HCV and/or HIV infection and their ambiguous nature in antiviral responses. 

As in prior studies by Rallon et al. [28], there was no observed correlation between regulatory T cells and activation status of CD4^+^ cells in patients with HCV alone and coinfected with HIV. However, as suggested recently [35], this does not exclude the possibility that depletion of T_regs_ may result in recovered responses of T cells affected by viral infection.

Despite the decreased number of regulatory T cells that may inhibit HCV-specific responses, we have found that pegIFN-α/RBV therapy significantly enhances the HIV-induced decline in absolute CD4^+^ T cells number. These observations are supported by prior work that shows a significant decline in the number of CD4^+^ T cells as a result of IFN-α/RBV treatment in HIV-infected participants [36]. Therefore, we propose that pegIFN-α/RBV treatment should be reconsidered in those with HCV/HIV co-infection. Interestingly, in recent years, an inverse correlation was found between T_reg_ frequency and CD4^+^ T cell counts in HIV-infected patients [13,37], and this was confirmed in our study amongst HCV/HIV-coinfected individuals. In addition, it is worth noting that previous studies have suggested an HCV-induced, apoptosis-mediated decline in CD4^+^ T cells in HCV/HIV-coinfected patients [16]. Thus, further investigation is required to evaluate the exact role of T_reg_ in HCV and/or HIV infection to explain the pronounced decline in CD4^+^ T cells counts in response to pegIFN-α/RBV, despite the concomitant decrease in regulatory T cell numbers.

We did not find any association between regulatory T cell counts and biochemical markers of liver damage, namely, AST, ALT, or GGT, in either studied group of patients; even in HCV-related liver dysfunction, despite the increased T_reg_ level, no such association has been found [38]. However, the significantly elevated APRI scores in HCV/HIV-coinfected patients correlated with CD4^+^CD25^+^CD127^−^ regulatory T cell counts, thus indicating an association between higher numbers of these cells and APRI-based predictions of hepatic fibrosis. This observation is consistent with prior data indicating elevated cell counts in HCV patients with HIV co-infection [39]. The analysis of regulatory T cells including the Foxp3 marker showed no correlation between Tregs and fibrosis-predicting factor (APRI), which is consistent with observations in the existing literature [14]. In addition, recent studies using mouse models indicate that depletion of cells with a regulatory phenotype may lead to liver fibrosis [21]. Moreover, patients with HCV/HIV-coinfection have been demonstrated to exhibit decreased numbers of hepatic Foxp3^+^ regulatory T cells in parallel with increased expansion of cytotoxic CD8^+^ cells when compared to individuals with HCV monoinfection [32]. Therefore, it is important to precisely evaluate the role of T_regs_ depending on their peripheral or tissue localization.

Previous data showed that increased values of Foxp3^+^ cells within CD4^+^CD25^+^ intrahepatic lymphocytes are associated with increased HCV viral load and accelerated disease progression [29,30]. Interestingly, in our study, a correlation between peripheral blood CD4^+^Foxp3^+^ T_reg_ and HCV viral load was found in HCV-infected patients; this negative association was observed in HCV/HIV-coinfected patients in reference to putative CD4^+^CD25^+^CD127^−^ regulatory T cell absolute numbers. Thus, higher levels of putative Tregs might explain the lower HCV viral load present in blood of HCV/HIV-coinfected patients compared to those infected with HCV alone. In addition, the positive correlation of CD4^+^Foxp3^+^ Tregs and HIV viral load in HCV/HIV-coinfected subjects supports the previously suggested hypothesis that HIV persistence might be supported by increased levels of cells with a regulatory phenotype [28]. In fact, Zhuang et al. also demonstrated a positive correlation between Tregs and HIV viral load in HIV-infected individuals [13]. In addition, it is worth noting that ribavirin alone is not singularly responsible for blocking HCV replication. Combination with pegIFN-α was demonstrated to enhance RBV effects even in in vitro conditions with no additional activity of the immune system involved [40]. Thus, the role of IFN-α as an essential element in the phenomenon observed in our study was justified. We agree with the previously suggested hypothesis that regulatory T cells may have two equally crucial roles in HCV infection: conferring virus persistence through suppression of effector T cells and prevention of excessive immunopathological reactions in intense antiviral responses [41].

Recent studies demonstrated that DAA reduces cytokines associated with liver damage, including pro-inflammatory TNF-α, IL-1β, and IL-6, with a concomitant decline in HCV viral load [42]. Interestingly, here we found that treatment implementing pegIFN-α/RBV induces higher levels of Tregs that also correlated positively with virus clearance. Therefore, it seems highly probable that DAA and pegIFN-α/RBV therapeutic regimens lead to similar immunological phenomenon involving participation of both pro-inflammatory and immunosuppresive factors. Regarding beneficial effects of DAA on liver function, in the course of anti-viral therapy reported by Sasaki et al. [42], despite an increase of Tregs after pegIFN-α-based treatment, no association has been shown in reference to liver related parameters—ALT, AST, and APRI. In accordance with that, it seems crucial to carefully monitor immunological and virological aspects of the patients when choosing between DAA or pegIFN-α/RBV therapy to achieve the highest therapeutic efficacy.

Despite the shift from IFN-based to DAA therapeutic protocols, there are still numerous data suggesting beneficial effects of pegIFN-α/RBV, especially in patients with HIV coinfection requiring such an approach. In accordance with data presented here, slight differences have been demonstrated in the achievement of a sustained viral response depending on the HCV genotype—the highest rates were obtained in types 1, 2, and 4 [43]. Interestingly, these genotype-related changes in Treg levels were not present in HCV-HIV-coinfected patients. We reported higher initial frequencies of Tregs in that group; however, it was not associated with a differential response to the treatment applied. Assuming that higher levels of Tregs in those with HCV genotype 3 in the course of pegIFN-α/RBV treatment contribute to the effective regulation of anti-viral responses, our data might support previously reported high SVR rates in these groups [44]. The demonstrated high efficacy of IFN-β in type 3 HCV (predominantly) was also evidence for the constant need of IFN-based approaches in certain groups of HCV-infected patients [45]. Cumulatively, in the world of DAA, there is still a place for interferons application in HCV therapy of certain patient groups, but only with well-established treatment selection criteria.

In summary, we found that the application of pegIFN-α/RBV therapy in HCV-infected patients was associated with the increased expansion of T cells with a regulatory phenotype, with prolonged elevated levels of these cells in HCV/HIV-coinfected individuals. Our data indicate that regulatory T cells may be a crucial component of the antiviral response, especially in HCV-infected patients. Considering the lack of change in regulatory T-cell phenotype and its correlation with HIV and HCV viral load, we presume that HIV-coinfection inhibits the expansion of Tregs in response to treatment in HCV/HIV-coinfected patients, therefore controlling virus persistence and HCV-related liver damage. However, further studies directed at regulatory T cells in HCV and/or HIV infection are needed to determine their specific role in the antiviral immune response.

## Figures and Tables

**Figure 1 viruses-13-01448-f001:**
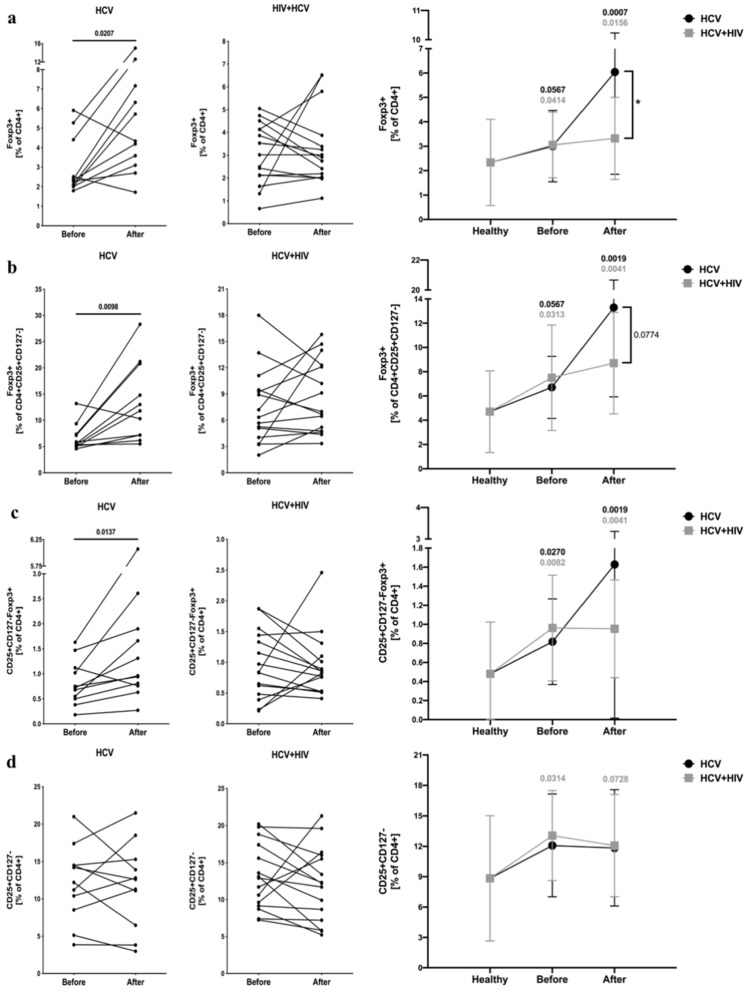
Frequencies of regulatory T cells in HCV-infected (left column) and HCV/HIV-coinfected (right column) patients in the course of anti-viral therapy. Data presenting the effects of therapy and differences between studied groups in the course of treatment in relation to healthy control subjects. Treg-related parameters included: Foxp3-positive cells within CD4^+^ (**a**) and CD4^+^CD25^+^CD127^−^ lymphocytes (**b**), frequency of CD25^+^CD127^−^Foxp3^+^ (**c**) and CD25+CD127- (**d**) in pool of CD4^+^ T cells.

**Figure 2 viruses-13-01448-f002:**
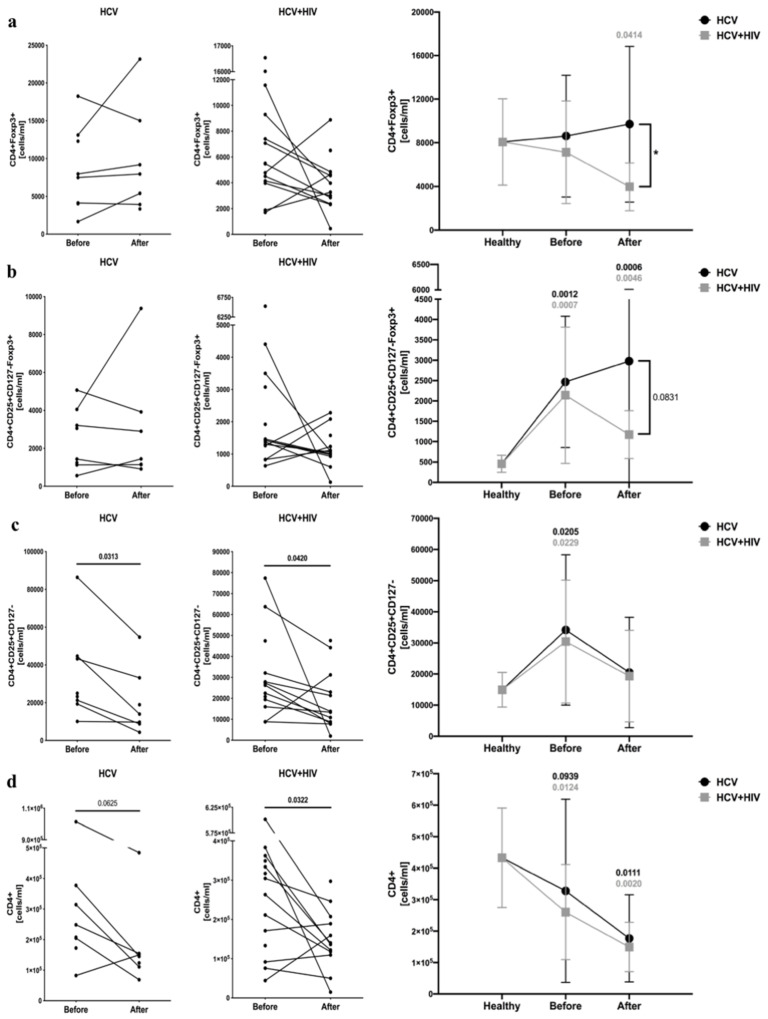
Anti-viral treatment effects on regulatory T cells absolute numbers in HCV-infected (left column) and HCV/HIV-coinfected (right column) patients. Data presenting the effects of therapy and differences between studied groups in the course of treatment in relation to healthy control subjects. Treg-related parameters included: CD4^+^Foxp3^+^ (**a**), CD4^+^CD25^+^CD127^−^Foxp3^+^ (**b**), CD4^+^CD25^+^CD127^−^ (**c**), and total CD4^+^ (**d**) lymphocytes.

**Figure 3 viruses-13-01448-f003:**
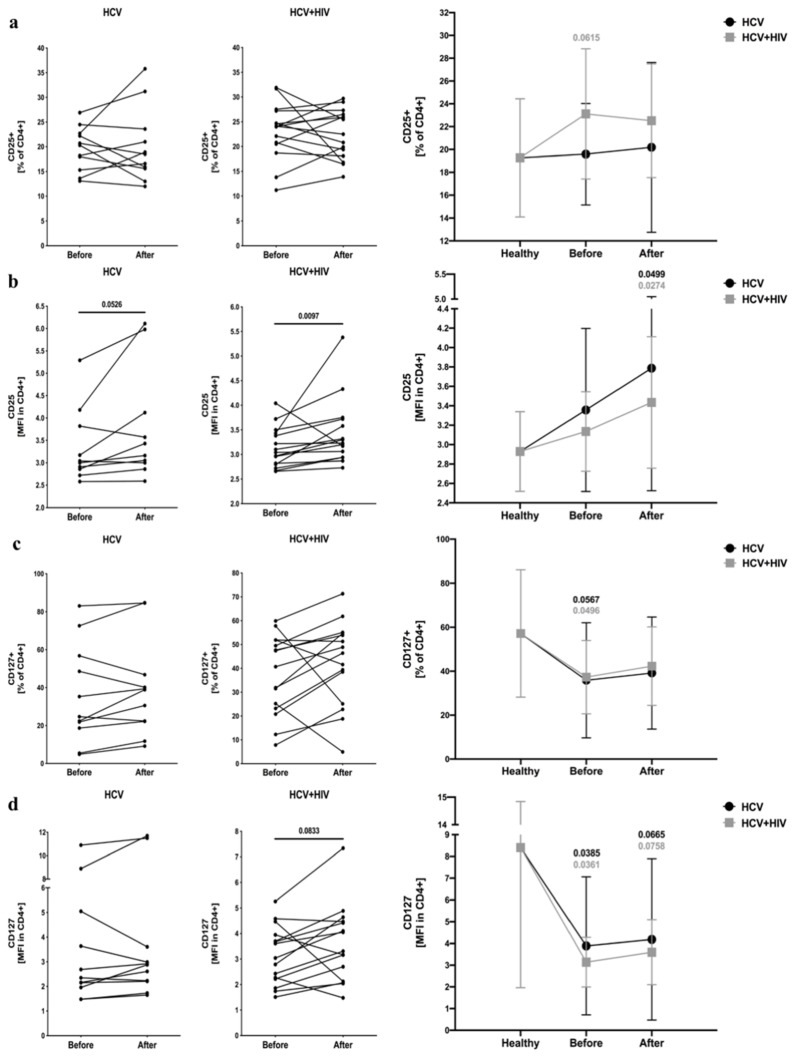
Alterations in T cells activation- (CD25) and development-related (CD127) proteins in the course of anti-viral treatment of HCV-infected and HCV/HIV-coinfected patients. Data presenting effects of therapy and differences between studied groups in the course of treatment in relation to healthy control subjects. CD25 activation marker was analyzed as frequency (**a**) and mean fluorescence intensity (MFI) (**b**) within CD4^+^ lymphocytes, and similarly, CD127 was demonstrated as frequency of marker-positive cells (**c**) and MFI (**d**) in CD4^+^ T cell pool.

**Figure 4 viruses-13-01448-f004:**
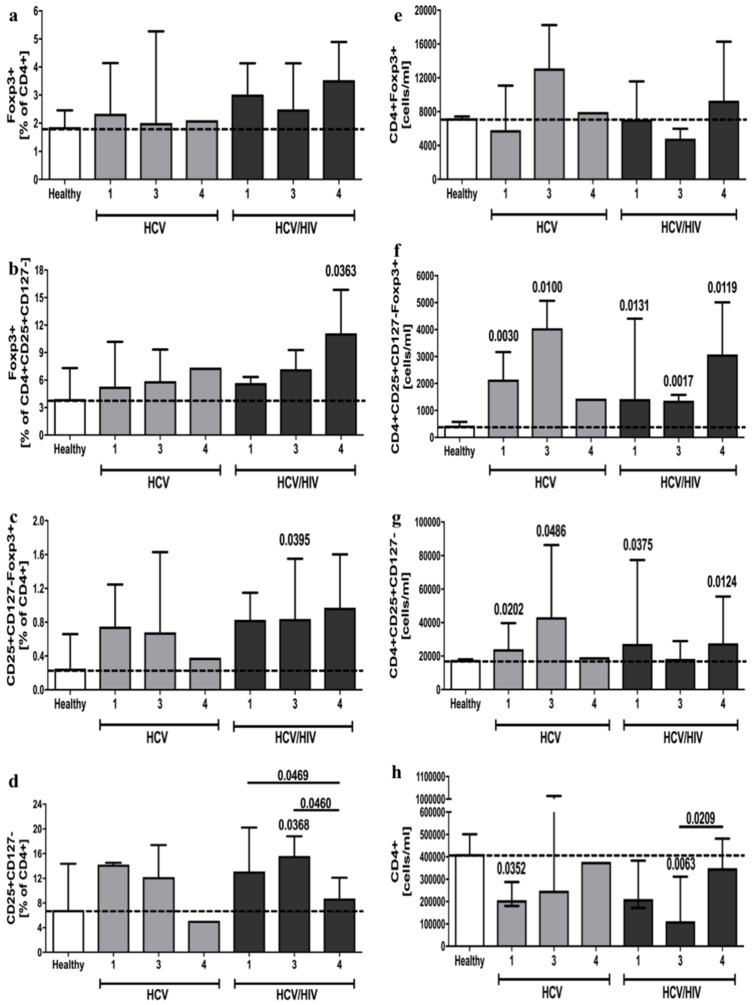
HCV virus genotype effects on frequency and absolute numbers of regulatory T cells in HCV-infected and HCV/HIV-coinfected patients. Treg-related parameters analyzed in context of frequencies: Foxp3^+^ in CD4^+^ (**a**) and CD4^+^CD25^+^CD127^−^ T cells (**b**), CD25^+^CD127^−^Foxp3^+^ (**c**) and CD25+CD127- (**d**) within CD4^+^ T cell pool; and absolute numbers: Foxp3^+^ (**e**), CD25^+^CD127^−^Foxp3^+^ (**f**), CD25+CD127- (**g**) CD4^+^ lymphocytes, and total CD4^+^ cells (**h**). Significant changes in relation to healthy controls indicated directly above specific group column; additional differences distinguished with lines connecting significantly different groups.

**Figure 5 viruses-13-01448-f005:**
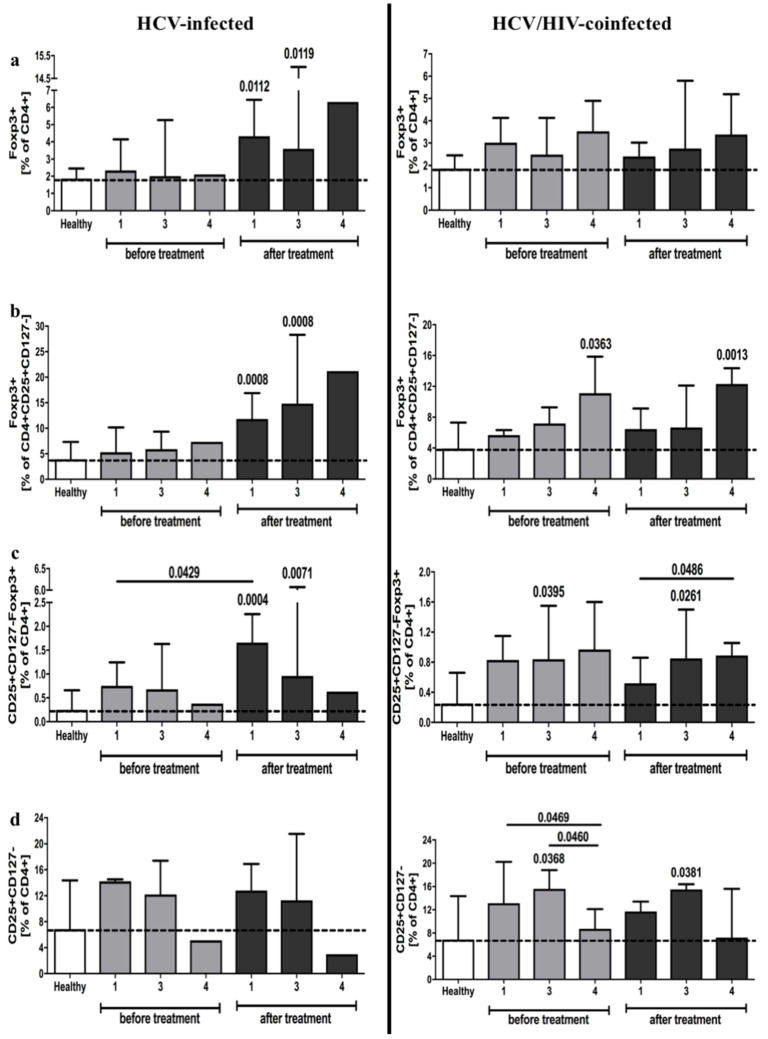
HCV virus genotype-related differences in regulatory T cells frequencies in response to anti-viral therapy in HCV-infected (left column) and HCV/HIV-coinfected patients (right column). Data related to T regs in HCV or HCV/HIV-coinfected patients presented as: Foxp3^+^ in CD4^+^ (**a**) and CD4^+^CD25^+^CD127^−^ (**b**) lymphocytes, CD25^+^CD127^−^Foxp3^+^ (**c**) and CD25+CD127- (**d**) within CD4^+^ T cell pool. Significant changes in relation to healthy controls indicated directly above specific group column; additional differences distinguished with lines.

**Figure 6 viruses-13-01448-f006:**
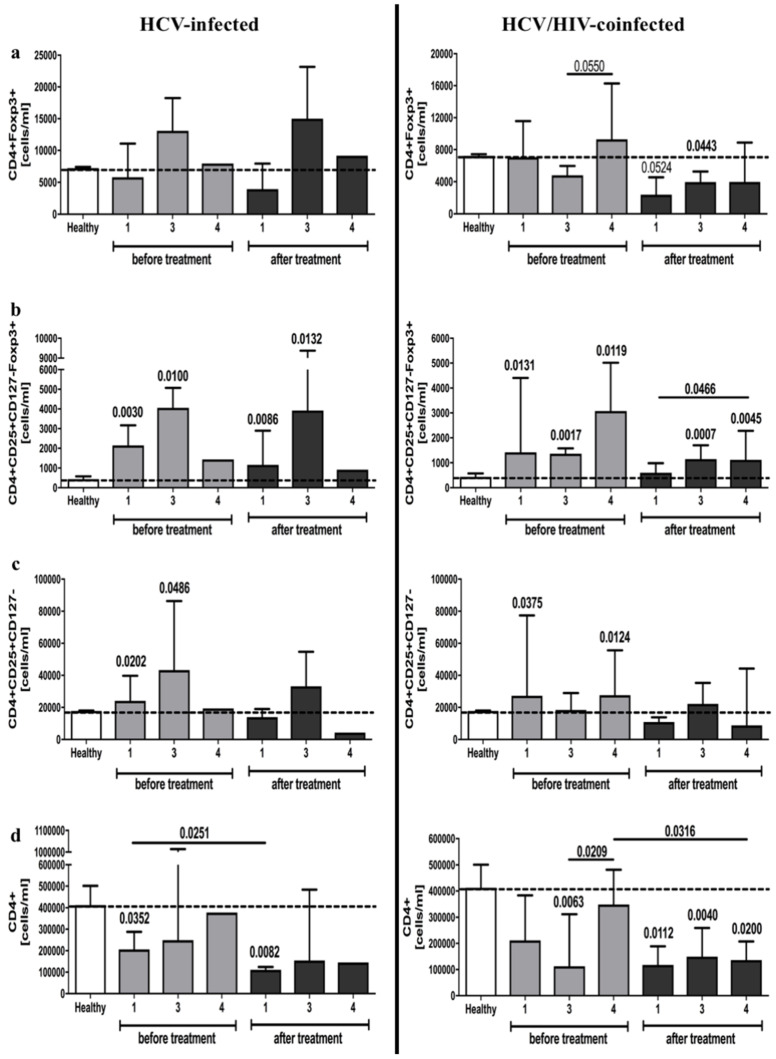
HCV virus genotype-related differences in regulatory T cells absolute numbers in response to anti-viral therapy in HCV-infected (left column) and HCV/HIV-coinfected patients (right column). Treg-related parameters in HCV or HCV/HIV-coinfected patients presented as absolute numbers: Foxp3^+^ (**a**), CD25^+^CD127^−^Foxp3^+^ (**b**), CD25+CD127- (**c**) CD4^+^ lymphocytes, and total CD4^+^ cells (**d**). Significant changes in relation to healthy controls indicated directly above specific group column; additional differences distinguished with lines connecting significantly different groups.

## Data Availability

The data presented in this study are available on request from the corresponding author. The data are not publicly available due to privacy restrictions.

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
