# Peer review of "Effects of Pegylated Interferon Alpha and Ribavirin (pegIFN-α/RBV) Therapeutic Approach on Regulatory T Cells in HCV-Monoinfected and HCV/HIV-Coinfected Patients"

_viruses, 2021, doi:10.3390/v13081448_

Round 1
Reviewer 1 Report
Grubezak et al. reported that Peg-interferon plus ribavirin treatment induces significant effect on regulatory T cells in the course of HIV and/or HCV infection, indicating that these play a role in antiviral response. Manuscript seems well written.
- In DAA era, enhanced host immune status may play an additive role on HCV RNA clearance by DAA (Sasaki R, et al. J Med Virol. 2019 Mar;91(3):411-418. doi: 10.1002/jmv.25310.). Please discuss about the difference between interferon-free treatment and interferon-including treatment.
- Authors also used ribavirin. How the role of ribavirin? Authors should discuss more (Kanda T, et al. J Viral Hepat. 2004 Nov;11(6):479-87. doi: 10.1111/j.1365-2893.2004.00531.x.).
Author Response
"Please see the attachment."

Reviewer 2 Report
In their submission, Effects of pegylated interferon alpha and ribavirin (pegIFN-α/RBV) therapeutic approach on regulatory T cells in HCV-monoinfected and HCV/HIV-coinfected patients, Grubczak et al. explore different Treg phenotypes in HCV monoinfected and HCV/HIV coinfected patients before and after treatment with pegIFN-a/RBV therapy. They found that pegIFN-a/RBV administration was associated with higher levels of Tregs only in HCV monoinfected patients. HCV/HIV coinfection seemed to inhibit such expansion of Tregs suggesting that HIV infection and/or HIV-ART treatment might contribute to this limited expansion.
I recommend that this manuscript be accepted after major revision.
Comments:
1) I recommend the authors to clarify in Material and Methods-Patients whether HCV/HIV coinfected patients were treated with HIV-ART treatment or not during the study and indicate which combined treatment was administered.
2) In Material and Methods-Patients, which was the limit of detection for plasma HIV RNA measurement?
3) In Material and Methods-Statistical analysis, the authors need to include how the correlation coefficients were determined.
4) In Results-Characteristic of the studied subjects based on immunological, biochemical and virological laboratory results, the authors point that HCV-infected patients seemed to have well-controlled HCV viral load, but a median value of 17.100 seems too high to be considered well-controlled. Did pegIFN-a/RBV administration result in lower HCV viral loads in both groups of patients? I recommend the authors to include extra columns in Supp. Tab. 1 and present immunological, biochemical and virological laboratory results after treatment as well.
5) In Results- Alterations in frequencies and absolute numbers of regulatory T cells in HCV-infected and HCV/HIV-coinfected patients in the course of anti-retroviral therapy, the authors mentioned “pegIFN-α/RBV administration in HCV-infected patients led to a significant decrease in total number of CD4+ T cells. In HCV/HIV-coinfected patients” but the p-value given is not significant, this should be noted as a slightly decrease or tendency.
6) In Results- Association between regulatory T cells and immunological, biochemical and virological laboratory parameters, in HCV-infected and HCV/HIV-coinfected patients. In order to support author’s conclusions, I recommend the authors to include as well associations found to be significant after pegIFN-α/RBV administration.
7) In Discussion, the authors need to include some discussion regarding the influence of HCV genotype on the differential response of Tregs to pegIFN-α/RBV administration. Were these results expected?
Author Response
"Please see the attachment."

Round 2
Reviewer 2 Report
The authors have addressed all my concerns and therefore I support publication without further changes.